# Effects of Aging on the Physical and Rheological Properties of Trinidad Lake Asphalt Modified Bitumen

**DOI:** 10.3390/ma14102532

**Published:** 2021-05-13

**Authors:** Krzysztof Kołodziej, Lesław Bichajło, Tomasz Siwowski

**Affiliations:** Department of Roads and Bridges, Faculty of Civil and Environmental Engineering and Architecture, Rzeszow University of Technology, 35-959 Rzeszow, Poland; krzych@prz.edu.pl (K.K.); leszbich@prz.edu.pl (L.B.)

**Keywords:** aging, bituminous binder, modification, Trinidad Lake Asphalt, physical properties, rheological properties

## Abstract

The application of various modifiers has emerged in recent years to improve conventional petroleum-based bitumen properties. The natural asphalt called Trinidad Lake Asphalt (TLA) has been applied very often due to its consistent properties, high viscosity and density, and superior rheological properties, and effective blending with other bitumen. However, most studies on TLA-modified binders always focused on physical and rheological properties in the original (unaged) condition, but the details about aging properties are often neglected. This study aimed to investigate the effect of short-term aging on the physical and rheological characteristics of the 35/50 base bitumen modified by the addition of two different TLA contents. The conventional physical tests and dynamic shear rheological tests were undertaken before and after aging to measure the penetration and softening point, complex shear modulus, and phase angle of the modified binders, as well as to calculate the zero shear viscosity using the Cross model fitting procedure. Based on the results of the above-mentioned comprehensive testing, the effect of aging on TLA-modified binder properties was evaluated using aging indices, as well as a direct comparison of results. The tests revealed that the short-term aging of TLA-modified binders did not worsen or reduce the pavement resistance to permanent deformation or the load-bearing capacity of the asphalt mixture.

## 1. Introduction

To improve conventional petroleum-based bitumen properties due to the increased demand for longer pavement service life, the application of various modifiers has emerged in recent years as a viable solution [1,2,3,4,5,6]. Among these modifiers, the natural asphalt called Trinidad Lake Asphalt (TLA) has been applied very often as a bitumen modifier. It consists of a semi-solid emulsion of soluble bitumen, mineral matter, and other minor constituents. TLA is well known for its consistent properties, high viscosity and density, superior rheological properties, and effective blending with other bitumen [7,8]. Furthermore, TLA has good temperature stability (low penetration and high softening point), chemical stability (good oxidation resistance), and water stability (good adhesivity), as well as good compatibility with petroleum bitumen due to their similar chemical compositions [9,10]. However, TLA is too hard to be used alone in bituminous mixtures; therefore, it is generally mixed with petroleum-based bitumen in certain blending ratios. TLA can be easily added to any bituminous mixture, in any quantity, at any time, and mixed at elevated temperatures with no degradation or loss of performance. It is routinely used in asphalt concrete (AC) and mastic asphalt (MA) pavements. In recent years, TLA has been particularly well used in modern thin bridge pavement systems [11,12,13,14].

TLA-modified binders present increased stiffness, mechanical strength, elastic modulus, and tensile stress; enhanced Marshall stability and adhesion; less tendency toward thermal susceptibility; lower solubility in solvents; less resistance to low-temperature cracking [9,15,16,17,18,19]. Many researchers investigated the rheological properties of TLA-modified binders [8,9,16,17,20,21]. Their results indicated that the addition of TLA into base bitumen significantly improved its rutting resistance, which is usually related to high-temperature performance. Furthermore, Ni et al. [22] revealed that the TLA addition enormously enhanced the fatigue pavement performance. In turn, Feng et al. [10] and Cao et al. [13] showed excellent high-temperature stability, the capacity for moisture damage resistance, and the impermeability of TLA-modified binders in their research. Summing up, TLA as a bitumen modifier improves pavement strength, anti-rutting performance, fatigue performance, durability, and life-cycle pavement costs (LCA). However, the improvement effectiveness is always case-sensitive and depends on the type of base bitumen, the weight percentage of TLA content in the bitumen, and the type of pavement (AC, MA). Moreover, since the cost of TLA is quite high, it is usually necessary to optimize the amount of TLA content in bitumen.

Most studies on TLA-modified binders always focused on physical and rheological properties in the original (unaged) condition, but the details about aging properties are often neglected [7,9,17,20]. Aging causes several changes in a binder as well as the bituminous mixture properties, which are reflected in the performance of the final pavement. Results from many studies showed that the penetration and ductility of binders are reduced while the softening point is increased as a result of aging. Ultimately, the viscosity of the binder is increased and becomes a stiffer bituminous mixture. Regarding binders’ mechanical properties, the stiffness modulus also increases due to aging. This may cause the mixture to become excessively hard and brittle and susceptible to disintegration and thermal cracking at low temperatures. Aging may also render the mixture less durable than the original mixture in terms of wear resistance and moisture susceptibility. As a result, the damage tolerance of bituminous pavements decreases, which can lead to premature failures in service. On the other hand, thanks to the stiffening effect of the short-term aging, the resistance of the original asphalt mixture to permanent deformation and the load-bearing capacity are not worsened or reduced due to aging.

There are only a few research papers on the aging properties of TLA-modified bitumen. Iwański et al. [23] carried out a comparative feasibility study of the influence of modern modifiers (TLA, Gilsonit, synthetic wax, adhesive agent) for incorporating them into the bituminous mixture at low production temperatures. The bitumen of penetration grade 35/50 was used as a reference binder. The authors revealed that short-term aging significantly influenced the bitumen characteristics and the highest bitumen change rate after aging was observed in the case of the use of natural bitumen. Li et al. [24] revealed that the addition of TLA can effectively improve the thermal-oxidative and UV aging resistance because the retained penetration increased while the softening point increased and the viscosity aging index decreased after an aging simulation of the TLA-modified binders. Xu et al. [21] showed that the influence of TLA on the short-term aging performance of bitumen is slight, but TLA enhances the long-term aging resistance, which is favorable for extending the lifetime of pavement. Słowik and Bilski [25] revealed that the unfavorable impact of the aging process on the TLA-modified binder properties is limited if a suitable TLA content is used; they propose the optimal TLA content of 15% by weight to modify the 35/50 and 50/70 penetration grade bitumen. Fengler et al. [8] demonstrated that, as the TLA content and the aging time increased, the binder showed a tendency to present accentuated cracking. Furthermore, the stiffness considerably increased after long-term aging when using the 50% TLA-modified binders, which led the material to exhibit block cracking. Bin et al. [26] found that the addition of TLA natural asphalt improved the anti-aging properties of the base asphalt. They used the complex modulus to model and predict the rheological properties of the bitumen under various aging conditions (aging time and temperature). However, despite some of the studies mentioned above, it is still very important to characterize the aging performance of TLA-modified binders to enlarge their application scope.

In the current research, the effect of aging on the physical and rheological properties of TLA-modified binders was evaluated and discussed in the context of using TLA-modified binder in a mastic asphalt (MA) mixture. The MA has been traditionally known as a durable and deformation-resistant bridge pavement with good fluidity and self-leveling performance when laying down. The mechanical properties of MA are highly affected by the properties of the respective binder and TLA is often used for modifying the binder properties to obtain improved performance of the respective MA mixture. Thus, the appropriate properties of the TLA-modified binder are crucial for obtaining durable MA bridge pavement. In this paper, the partial results of the domestic research on MA properties [27], as a widely used and specified material for bridge pavements, are presented.

This study aimed to investigate the effect of short-term aging on the physical and rheological characteristics of the 35/50 base bitumen modified by the addition of two different TLA contents. At first, the conventional physical tests evaluating the penetration and softening point were carried out before and after aging. Furthermore, dynamic shear rheological (DSR) tests were undertaken before and after aging to measure the complex shear modulus and phase angle of the modified binders, as well as to calculate zero shear viscosity using the Cross model fitting procedure. The rolling thin film oven test (RTFOT) was used to simulate short-term aging. Since short-term aging has a significant influence on the resistance of hot mix asphalt regarding permanent deformation, while long-term aging affects the resistance to fatigue (not important for MA pavements), only short-term aging test results were considered in this part of the research [23,24,25]. Based on the results of the above-mentioned comprehensive testing, the effect of aging on the TLA-modified binder properties was evaluated using aging indices, as well as a direct comparison of results.

## 2. Materials and Methods

### 2.1. Materials Properties

Since the 35/50 penetration-grade bitumen is the only one recommended for MA bridge pavements by the Polish road administration, it was chosen as the base bitumen in this study. Moreover, the 35/50 bitumen is also the most popular one in Poland for use in other types of road pavements. Based on the available studies [9,20,21,25] and the authors’ initial research [27], the TLA content by bitumen weight up to 20% was chosen to modify the base bitumen properties toward the durable MA applications on bridges. Due to the high cost of the TLA modifier, the chosen TLA content was also optimized based on the life-cycle pavement cost analysis (LCCA). Since the TLA-modified binders to be used in MA pavement were evaluated in this study and MA is a fatigue-resistant mixture due to its high binder content, the long-term aging properties were not considered at this stage of research.

Two combinations of the 35/50 penetration-grade bitumen (base bitumen) and the TLA were selected to evaluate the aging characteristics of TLA-modified bitumen. Table 1 lists some of the basic properties of the 35/50 bitumen in the original (unaged) condition, which were determined according to the relevant European standards and compared with the European requirements.

The TLA used in this study was Trinidad Epure TE Z 0/8 (Trinidad Lake Asphalt, Bremen, Germany; PL Partner: CMS Polska sp. z o.o., Gdynia, Poland), whose basic properties are listed in Table 2.

As shown in the relevant tables, the base bitumen used in this study fulfilled the respective European requirements. However, in the case of TLA, small deviations from the requirements were revealed. This was due to the fact that the natural asphalt supplied by the manufacturer was covered by diatomite (an agent that prevents asphalt pieces from sticking together) and is added to the mixer in this form. It was assumed in this study that the tests would be conducted on the binder delivered in the form available from the manufacturer. Hence, this form of TLA was used in the tests. This additive can stiffen the TLA, resulting in the revealed deviations.

### 2.2. Preparation of the TLA-Modified Binders

To obtain a homogeneous TLA-modified binder, the laboratory blender was used to ensure a constant mixing speed and thus no voids were created in the mixture. The base bitumen and TLA were heated until they became a fluid before mixing. The base bitumen was preheated at 160 °C in an oven for 0.5 h to make it ready to mix. The modifier, which was a specified amount of TLA, was added into the liquid base bitumen with the external addition method. The temperature of the modified binders was kept at 160 °C for 1 h, as recommended by the 35/50 bitumen manufacturer. After this time, the mixture was stirred at 3000 rpm for 5 min according to [34] such that the TLA was homogeneously dispersed in the base bitumen. Immediately after preparation, the ready mixture was poured into short-term aging containers.

### 2.3. Aging Procedure

The RTFOT test was conducted on the base bitumen and modified binders following the European standard [35]. This method consists of exposing a thin layer of modified binder to hot air for 75 ± 1 min. Glass bottles with the samples of the binder are placed in a special disc that rotates at 15.0 ± 0.2 rpm located in a laboratory oven, where hot air was periodically injected at a rate of 4.0 ± 0.2 L/min. The binder samples in the bottles were subjected to a temperature of 163 ± 1 °C.

After the short-term aging, the binder was poured into a large container, homogenized via mixing, and then poured into molds for the appropriate tests. Efforts were made to minimize additional heating of the binder after the simulated short-term aging in order to not additionally age it. The physical properties tests were performed immediately after the aging procedure. Before the rheological properties tests, the aged samples were poured into molds, left for 24 h, and then the tests were started.

### 2.4. Physical Properties Tests

Conventional tests to measure the penetration at 25 °C and the ring and ball (R&B) softening point were carried out to study the physical properties of the 35/50 base bitumen, as well as the modified binders, before and after RTFOT aging. The penetration test following the European standard [28] is a common test that is performed to characterize the hardness of bitumen and binders. The R&B softening point test was carried out following the European standard [29] to determine the temperature at which a phase change in the binder occurred. For both physical tests, the Grubbs test was performed to detect outliers in a univariate data set that was assumed to come from a normally distributed population.

### 2.5. Rheological Properties Tests

Sinusoidal oscillation loading was utilized to undertake dynamic shear rheological tests (DSRs). The DSR measures a sample’s complex shear modulus G* and phase angle δ. The first value can be considered the sample’s total resistance to deformation when shear loaded, while the latter is the lag between the applied shear stress and the resulting shear strain. Typically, the higher the G* value, the stiffer the binder is (it is able to resist deformation), and the lower the δ value, the greater the elastic portion of G* is (it is able to recover its original shape after being deformed by a load).

However, based on the SHRP specification, Bahia [36] suggests the use of these two parameters (G*, δ) at a fixed temperature and frequency of testing. Since the viscosity of the binder changes with temperature and the elastic properties of the binder change with the frequency of loading, the above parameters were suggested to be ineffective in predicting the rutting performance of binders, especially in the case of modified binders [37,38,39,40]. The rutting parameter G*/sinδ does not fully evaluate the delayed elasticity; this parameter measures a short loading time, which is representative of traffic loading. For some binders, this process is long, which leads to underestimating the resistance to permanent deformation. A parameter that includes both features is the zero shear viscosity (ZSV) [41]. The concept of ZSV was introduced by Sybilski [42] to evaluate the contribution of binders to the rutting resistance of bitumen pavements. ZSV is a suitable indicator for evaluating the partial contribution of the binder to the rutting resistance of bitumen pavement layers. Thus, in this study, the ZSV of the TLA-modified binders was calculated using the different models and test methods.

The available literature describes many test methods for the experimental determination of the ZSV of binders based on the use of the DSR under different experimental conditions, in both oscillatory and creep modes [43,44,45]. In this study, the ZSV was determined using two different test modes: frequency sweep test and creep test. Following the preparation of the modified binders, both rheological tests were conducted using the DSR for original (unaged) and RTFOT-aged samples. For both rheological tests, the Grubbs test was carried out to detect outliers in a univariate data set that was assumed to come from a normally distributed population. To calculate the ZSVs of the modified binders, the Cross model was fitted to measure the complex viscosity η*. This model allows for extrapolating the complex viscosity at a zero frequency condition using nonlinear regression analysis.

#### 2.5.1. Frequency Sweep Test

The dynamic oscillatory tests were performed according to the European standard [46] using a HAAKE RheoStress 1 Rheometer (Thermo Electron, Karlsruhe, Germany) under strain-controlled loading conditions by applying a sinusoidal angular displacement of constant amplitude. The test geometry with a 25 mm diameter plate and 1 mm gap was used. A constant strain of 5% was applied over a range of test temperatures (between 40 and 60 °C with increments of 10 °C) and frequencies (between 0.01 and 100 Hz). All tests were carried out in the linear viscoelastic range.

The tests were performed on smooth plates, they were not additionally roughened. Before testing, the plates were heated to a temperature of 70–80 °C to soften the binder and ensure good adhesion to the plates. After proper preparation of the sample, it was cooled down to the test temperature. Before the test, the linear viscoelastic interval was determined. The deformation level was selected such that the sample in the entire range of tested frequencies was within this range.

The Cross model was used to estimate a zero shear viscosity of the modified binders [47]. The Cross model describes flow curves of the modified binder in the form of a four-parameter function as follows:(1)$$\frac{\eta^{*}-\eta_{\infty }^{*}}{\eta_{0}^{*}-\eta_{\infty }^{*}}=\frac{1}{1+{\left(K\omega \right)}^{m}}\;$$
where:
$\eta^{*}$ is the complex viscosity;$\eta_{0}^{*}$ is the ZSV;$\eta_{\infty }^{*}$ is the limiting viscosity in the second Newtonian region;ω is the angular frequency (rad/s);*K* and *m* are constants.


The HAAKE RheoWin (ver. 4.41, Thermo Electron, Karlsruhe, Germany), which is a complete measuring and evaluation software package, was used to control the rheometer and to handle the measured data with the selected rheological evaluation method. In this study, the Cross model material parameters ($\eta_{0}^{*}$, $\eta_{\infty }^{*}$, *K*, *m*) were determined using this software.

#### 2.5.2. Creep Test

The ZSV ($\eta_{0}^{*}$) can be also obtained from a simple creep test if the load is applied until a steady state occurs. The ZSV in the creep mode is typically determined using a rheometer in the temperature domain. In this rheological test, the constant direct shear stress is applied to a sample and the resulting deformation is measured as a function of loading time. The disadvantage of this method is that it takes a considerable amount of time. The viscosities were calculated via Equation (2) using the Burgers model [48]:(2)$$\eta_{i}=\frac{\Delta t}{\Delta J}=\frac{900}{J_{end}-J_{15\min before\;end}}\;$$
where:
$\eta_{i}$ is the steady-state viscosity (SSV) (Pa·s);Δt is the observation period (s);ΔJ is the compliance difference over the last 15 min (1/Pa);$J_{end}$ is the compliance measured at the end of the creep test (1/Pa);$J_{15\min before\;end}$ is the compliance 15 min before the load is removed (1/Pa).


In the creep method, the ZSV was determined according to the CEN standard [49] at the specified test temperatures: +40 °C, +50 °C, and +60 °C. The same test geometry with a 25 mm diameter plate and 1 mm gap was used. The test consisted of two steps: in the first step, a stress sweep was executed to select the creep test stress, and in the second step, the creep test was performed with the selected stress for a sufficiently long time to reach a steady-state flow. The applied shear stress was τ = 50 Pa.

### 2.6. Aging Indices

The aging properties of bitumen can be evaluated using aging indices, namely, the ratio or difference value of a performance parameter of the aged bitumen to that of the unaged one. The aging indices adopted in this study were obtained from the measurement of physical and rheological properties of base bitumen and TLA-modified binders before and after RTFOT aging. The following indices were calculated and evaluated: retained penetration (RP) at 25 °C, softening point increment (SPI), complex modulus aging index (CAI), phase angle aging index (PAI), aging index (AI), and viscosity aging index (VAI). They are defined by the following Equations (3)–(8):(3)$$RP=\frac{Penetration\;of\;aged\;binder}{Penetration\;of\;unaged\;binder}\times 100\%\;$$
(4)SPI=Aged softening point−Unaged softenig point 
(5)$$CAI=\frac{Aged\;complex\;modulus}{Unaged\;complex\;modulus}\;$$
(6)$$PAI=\frac{Aged\;phase\;angle}{Unaged\;phase\;angle}\;$$
(7)$$AI=\frac{Aged\;viscosity\;value}{Unaged\;viscosity\;value}\;$$
(8)$$VAI=\frac{Aged\;viscosity\;value-Unaged\;viscosity\;value}{Unaged\;viscosity\;value}$$

Higher values of *SPI*, *CAI*, *VAI*, and *AI* reflect a more significant aging influence. Conversely, lower values of *RP* and *PAI* reflect a more serious aging effect.

## 3. Results and Discussion

### 3.1. Physical Properties

TLA had low penetration and a high softening point due to the high content of ash and asphaltenes; therefore, the addition of TLA had a remarkable influence on the properties of the base bitumen. The physical properties determined for the TLA-modified binders before and after aging are presented in Table 3 and the following Figure 1, Figure 2 and Figure 3.

Figure 1a presents a substantial reduction in penetration value after the RTFOT aging compared to the unaged bitumen. The decrease in the penetration value was caused by the stiffening of the binders as a result of the aging processes. An increase in the binder stiffness arising from short-term aging is comparable for both modified binders discussed and fell within the range of 67.2 to 70.1%. The penetration before and after the RTFOT aging had a similar variation (decrease) with the increasing TLA content. When the TLA content was up to 20%, the penetration of the RTFOT-aged binder rapidly decreased, as in the case of the unaged one. Figure 1b shows the RP index of the base bitumen and TLA-modified binders. Generally, the TLA content showed no significant effect on RP. For both TLA-modified binders, the RP value decreased by about 6% and 3% for 10% and 20% TLA content, respectively, as compared to the base bitumen. This means that the addition of TLA can slightly worsen the short-term aging performance of TLA-modified binders.

The addition of TLA had a statistically significant influence on the obtained penetration values before (F(2, 15) = 201.76, *p* < 0.001) and after (F(2, 15) = 138.35, *p* < 0.001) the short-term aging. There was a statistically significant interaction between the TLA addition and aging: F(2, 30) = 13.34, *p* < 0.001. The effects of different TLA levels depended on what level of aging was present.

Figure 2a presents a substantial increase in the softening point temperature after the RTFOT aging compared to the unaged bitumen. It also shows that the softening point in both aging conditions, i.e., unaged and RTFOT-aged, changed with the TLA content. In both conditions, the softening point of the modified binders linearly increased with increasing TLA content. The addition of TLA slightly altered the softening point increment (SPI) as a result of aging, i.e., the average increment values were 6.6 °C for the base bitumen and 8.2–8.4 °C for the TLA-modified binders (Figure 2b). However, the rate of SPI was higher when the TLA content was equal to or larger than 10%. However, according to the European standard [31], the SPI for 35/50 bitumen may not be higher than 8.0 °C; the base bitumen fulfilled this requirement but the TLA-modified binders did not.

The addition of TLA had a statistically significant influence on the obtained softening point values before (F(2, 15) = 156.77, *p* < 0.001) and after (F(2, 15) = 370.90, *p* < 0.001) the short-term aging. There was a statistically significant interaction between the TLA addition and aging: F(2, 30) = 17.12, *p* < 0.001. The effects of different TLA levels depended on what level of aging was present.

As shown in Table 3, the addition of TLA had little influence on the short-term aging behavior of the base bitumen. According to the aging indices, the samples of 10% TLA and 20% TLA binders showed almost the same anti-aging level (a slightly better effect was found for 20% TLA), which means that the anti-aging performance of 35/50 base bitumen could not be improved continuously by increasing the TLA content. This suggests that the 20% TLA content seemed to be a little bit better concerning the aging performance of base bitumen.

### 3.2. Rheological Properties

#### 3.2.1. Complex Modulus and Phase Angle

The most important effect of a modifier is to enhance the viscoelasticity of the base bitumen because the shear resistance at high temperature has a strong correlation with the modulus of the bitumen. Table 4 and Figure 3, Figure 4, Figure 5 and Figure 6 present the results of the G* and δ determination at a frequency of 10 Hz for the unaged and RTFOT-aged conditions of the base bitumen and two modified binders evaluated at three different test temperatures, as well as the respective aging indices (CAI, PAI).

Figure 3 and Figure 4 reflect the rheological properties before and after aging. After the TLA-modified binder was aged using the RTFOT procedure, both the G* of the modified bitumen increased and the δ decreased consistently. Similar to the physical tests, this confirms that in the RTFOT-aged condition, the TLA-modified binders were stiffer than in the unaged condition in the entire frequency range. It can be observed in Figure 3 that the stiffness G* of the binders was directly proportional to the TLA content, both in the unaged and RTFOT-aged conditions. In contrast, the phase angle δ decreased proportionally to the TLA content in both conditions.

The temperature sweep test results are shown in Figure 4a. As shown in this figure, the G*’s of the modified binders were larger than that of the base bitumen at the specified test temperature, both for the unaged and RTFOT-aged conditions. Although the difference in G* between the diverse TLA-modified binders was relatively narrow, a trend where G* increased with the increase in TLA content also existed. When the test temperature was between 40 and 60 °C, the 20% TLA-modified binder had the strongest shear resistance because the G* was the largest, both for the unaged and RTFOT-aged conditions.

The addition of TLA had a statistically significant influence on the obtained stiffness modulus at different test temperatures: for test temperature 40 °C—F(2, 14) = 189.14, *p* < 0.001; for test temperature 50 °C—F(2, 15) = 121.64, *p* < 0.001; for test temperature 60 °C—F(2, 15) = 148.08, *p* < 0.001. There was a statistically significant interaction between the TLA addition and the test temperature: F(4, 44) = 131.06, *p* < 0.001. The effect of different TLA levels depended on what test temperature was present. The same conclusions could be drawn for the binder after the short-term aging: for test temperature 40 °C—F(2, 14) = 108.74, *p* < 0.001; for test temperature 50 °C—F(2, 13) = 159.50, *p* < 0.001; for test temperature 60 °C—F(2, 15) = 646.77, *p* < 0.001. There was a statistically significant interaction between the TLA addition and test temperature for the binder after the short term-aging: F(4, 42) = 62.97, *p* < 0.001.

The development of the δ of the base bitumen and the TLA-modified binders at the specified test temperature can be seen in Figure 4b. In contrast to the results of G*, δ decreased with the increase in TLA content within the temperature range. At the same temperature, the higher the TLA content, the smaller the δ of the TLA-modified binders. Furthermore, the bitumen with the lowest δ was the 20% TLA-modified binder, which reveals that the TLA modifier improved the elasticity of the base bitumen. This behavior was observed for unaged and aged binders. The δ of the bitumen and TLA-modified binders increased with the temperature increase, which indicates that the viscous property was enhanced with the temperature rising gradually, while the elastic recovery capacity decreased.

Summing up and considering G* and δ, both the TLA modification and RTFOT aging procedures enhanced the elastic response of a binder and increased the binder’s ability to resist deformation in the temperature range of 40–60 °C, demonstrating the effect of TLA and RTFOT in improving the thermal stability of the binder, which corresponds to the results of physical properties.

Figure 5 shows the Black curves for three studied binders in different aging conditions. As the binders aged, the rheological characteristics moved from the upper-right to the lower-left on the Black space diagram. The binders had similar initial properties; nevertheless, they appeared to resist deformation more effectively after the RTFOT aging.

The rheological aging indices (CAI and PAI) of the three binders with different TLA contents are presented in Figure 6. Considering the results after the RTFOT aging, both the TLA-modified binders exhibited smaller CAI and PAI within the entire sweep temperature range in comparison with the base bitumen. However, the influence of the TLA (i.e., better aging resistance) slightly depended on the test temperature. As far as the CAI was concerned, the best aging resistance of the TLA-modified binders was obtained at 40 °C, regardless of the TLA content. At the other temperatures, both worse aging resistance and TLA content dependence were shown. Moreover, at 60 °C, the 20% TLA content slightly reduced the aging resistance as compared to the base bitumen. Concerning the PAI, the bigger the TLA content, the smaller the PAI, i.e., worse aging resistance was observed, regardless of the test temperature. Both the CAI and PAI showed that the aging influence was more visible at higher test temperatures. Considering all of the analyzed properties (RP, SPI, CAI, and PAI), it can be observed that for 20% of TLA addition, three of the four properties were better than for the 10% addition (the CAI and PAI as the average values for temperatures 40 °C, 50 °C, and 60 °C).

#### 3.2.2. Zero Shear Viscosity

The ZSVs that were determined for the base bitumen and TLA-modified binders are presented in Table 5 and Figure 7, Figure 8, Figure 9, Figure 10, Figure 11 and Figure 12 depending on the test method (frequency sweep test or creep test), test temperature (40 °C, 50 °C, or 60 °C), and binder condition (unaged or RTFOT-aged). The respective aging indices (VAI, AI) are listed in Table 5 as well.

Figure 7 shows the curves for the complex viscosity vs. frequency for the base bitumen and two TLA-modified binders, each one in an unaged and RTFOT-aged condition. It was observed that, for each binder, the complex viscosity depended on the aging condition in the entire frequency range. The complex viscosity of the base bitumen and TLA-modified binders decreased with the faster shear frequency and increased with higher TLA content. Figure 7 also shows the fitting Cross curves for the three temperatures for the unaged and RTFOT-aged base bitumen and TLA-modified binders. Dot markers represent the experimental data and lines represent the fitting for each model. It was observed that the models properly describe the relationship between the complex viscosity and the frequency.

Table 6 shows the parameters for each model (obtained via least-squares fitting) and the evolution of each parameter with the aging condition of the binders. The ZSV values are listed in the third column (η_0_), and it was observed that this parameter depended on the TLA content and the aging condition. The short-term aging increased the ZSV values for all binders. Comparing the ZSV values for the three binders in their unaged and RTFOT-aged conditions, the 20% TLA-modified binder showed the largest ZSV value, followed by the 10% TLA-modified binder and base bitumen. The higher the TLA content, the larger the ZSV value was, which shows the influence of the TLA content on the complex viscosity of the 35/50 base bitumen.

In Figure 8, the creep curves for the base bitumen and the TLA-modified binders at the specific temperature are plotted. For each TLA content, the sample reached a steady state in the last 15 min of the test. Regardless of the aging condition, the addition of TLA increased the viscosity stiffening effect of TLA, which was revealed by much smaller deformations at the same shear stress τ = 50 Pa. The viscosity stiffening effect of the TLA depended on the specific test temperature, both for the unaged and RTFOT-aged binders. The higher the test temperature, the lower the viscosity stiffening effect of the TLA; the increase in creep deformation was seen at the end of the test. The shapes of the creep curves were similar regardless of the aging condition and TLA content, which revealed the stiffening effect of TLA, not the modification of the binder character itself. Generally, aging decreased the creep deformation of binders, regardless of TLA content level.

Same as for the frequency sweep test, in the creep test, the proportional decrease of the ZSV was observed with a decrease in TLA content and an increase in temperature, regardless of the aging condition (Figure 9). In the unaged condition, the twofold ZSV increase was obtained for 10% TLA content, and slightly less than a threefold increase for 20% TLA as compared to the base bitumen. At the remaining temperatures, the relevant ZSV increases were smaller: 1.7 and 2.5 times at 50 °C and 1.6 and 2.4 times at 60 °C, for 10% and 20% TLA, respectively. All the above-mentioned increases were slightly greater in the RTFOT-aged condition. Figure 10 shows the ZSV values obtained for both models and the dependence of the ZSV on the aging condition. It was observed that ZSV increased after RTFOT aging, then the resistance to permanent deformation also increased as the binder ages. The highest ZSV values were observed at 40 °C. The linear decrease of the ZSV could be observed as the test temperature rose for both unaged and RTFOT-aged binders (Figure 10).

The rheological tests performed in this study allowed for evaluating different methods for determining the ZSV of modified binders at low temperatures before and after aging. In Figure 11, the ZSV values obtained from the creep test were plotted versus the ZSV values predicted using the Cross model. A very good correlation was found between the creep method and the values calculated using the Cross model, which leads the authors to conclude that the Cross model fit the data very well and could be used reliably to estimate the ZSV, both in the unaged and RTFOT-aged conditions.

Figure 12 illustrates the viscosity aging indices: AI and VAI, which were calculated for each test temperature for the TLA-modified binders. The test method (frequency sweep or creep) was also considered in the latter case. A greater value of AI and VAI reflects a more significant influence due to aging. Both indices gradually decreased with the increase of test temperature but increased with the increase of TLA content. In particular, considerable AI and VAI increases were observed for the 20% TLA modified binder. In this case, both AI and VAI increased firstly and then get gradually declined. The influence of the test method is also revealed in Figure 12b. For 0% and 10% TLA content, the smaller VAIs were obtained in the creep test but the opposite dependence was obtained for the 20% TLA modified binder, where the creep values were much greater than those obtained in the frequency sweep test.

The test method did not result in obtaining statistically significant differences between the results of the ZSV tests for the binder before and after the short-term aging, i.e., t(16) = −0.156, *p* = 0.878 and t(16) = −0.0733, *p* = 0.942, respectively. As the ZSV test method did not significantly affect the result, further statistical analysis was performed for the frequency sweep test.

The addition of TLA had a statistically significant influence on the obtained ZSV at different test temperatures: for test temperature 40 °C—F(2, 15) = 320.85, *p* < 0.001; for test temperature 50 °C—F(2, 15) = 162.90, *p* < 0.001; for test temperature 60 °C—F(2, 15) = 217.07, *p* < 0.001. There was a statistically significant interaction between the TLA addition and the test temperature: F(4, 45) = 274.10, *p* < 0.001. The effects of different TLA levels depended on what test temperature was present. The same conclusions were drawn for the binder after the short-term aging: for test temperature 40 °C—F(2, 15) = 232.17, *p* < 0.001; for test temperature 50 °C—F(2, 15) = 685.97, *p* < 0.001; for test temperature 60 °C—F(2, 15) = 1561.00, *p* < 0.001. There was a statistically significant interaction between TLA addition and the test temperature for the binder after short-term aging: F(4, 45) = 200.27, *p* < 0.001.

## 4. Conclusions

A comprehensive laboratory investigation on the effects of short-term aging on the physical and rheological characteristics of 35/50 base bitumen that was modified via the addition of two different TLA contents (10% and 20% by weight) was carried out and presented in this paper. Based on the test results of this study, the following conclusions were made:
A substantial reduction in penetration value (RP = ~68.5%) and a substantial increase in softening point temperature (SPI = ~8.3 °C) were obtained after the RTFOT aging of both TLA-modified binders. The 20% TLA content is recommended regarding the aging performance of base bitumen.Similar to the physical tests, the rheological tests (G*, δ) confirmed that in the RTFOT-aged condition, the TLA-modified binders were stiffer than in the unaged condition over the entire frequency range. The stiffness of the binders was directly proportional to the TLA content, both in the unaged and RTFOT-aged conditions. Considering the G* and δ values obtained in the tests, both the TLA modification and RTFOT aging procedures increased the binders’ ability to resist deformation in the temperature range of 40–60 °C and enhance the elastic response of the binders, which corresponded to the changes in physical properties.The zero shear viscosity (ZSV) was used to evaluate the partial contribution of the binder to the rutting resistance of bitumen pavement. The bigger the ZSV, the better the elasticity of the binder and the smaller the permanent plastic deformation. The short-term aging increased the ZSV values for all binders and decreased the creep deformation of the binders, regardless of the TLA content level. The 20% TLA-modified binder showed the largest ZSV value. The Cross model fit the data very well and was reliably used to estimate ZSV, both in the unaged and RTFOT-aged conditions.Considering all the analyzed properties (RP, SPI, CAI, PAI, and ZSV) in the rutting resistance context, it was observed that for a 20% TLA addition, four of the five properties were better than for the 10% addition, but the test results suggested that the binder with 20% TLA was the most susceptible to short-term aging.


The aging of bituminous binders is one of the key factors determining the lifetime of an asphalt pavement. The tests revealed that the short-term aging of TLA-modified binders did not worsen or reduce the pavement’s resistance to permanent deformation or the load-bearing capacity of the asphalt mixture. The use of TLA greatly improved the high in-service properties of bitumen and enhanced the rutting resistance, as expected. On the other hand, the probable deterioration of the low-temperature characteristics is also possible because the TLA would give rise to a much harder bitumen at low temperatures and therefore a high tendency to undergo thermal cracking.

It should also be emphasized that this study is a part of broader research on the effect of TLA addition on the resistance of mastic asphalt mixture to permanent deformation. The obtained ZSV results will be used to evaluate mastic asphalt mixtures and to determine the correlations between TLA addition and parameters that defined permanent deformation.

## Figures and Tables

**Figure 1 materials-14-02532-f001:**
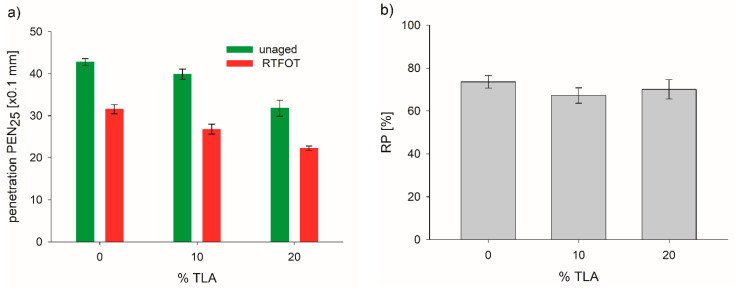
Penetration (**a**) and retained penetration index (**b**) before and after RTFOT aging of modified binders for different TLA contents.

**Figure 2 materials-14-02532-f002:**
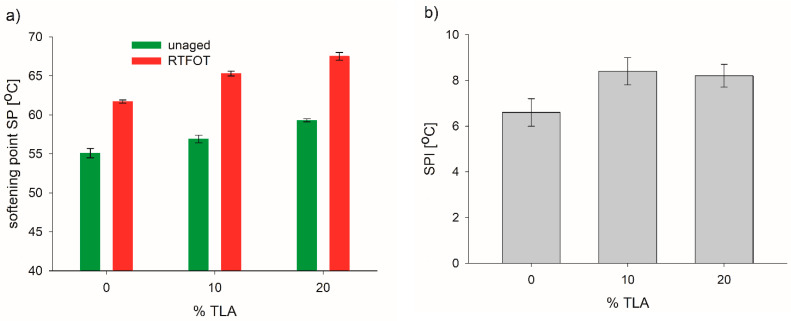
Softening point (**a**) and softening point increase (**b**) before and after the RTFOT aging of modified binders for different TLA contents.

**Figure 3 materials-14-02532-f003:**
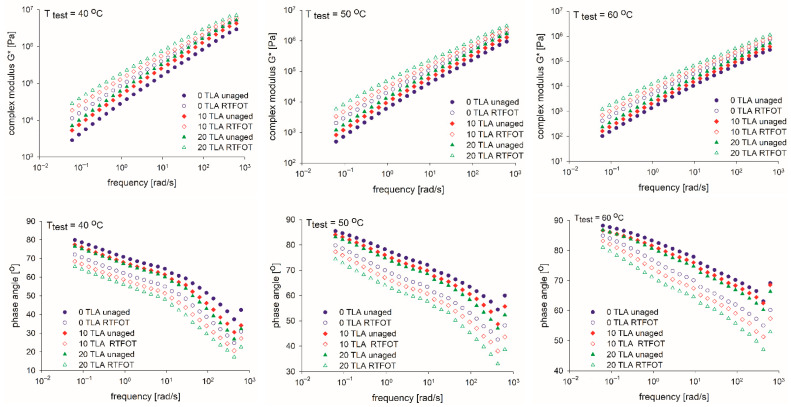
Isothermal curves of the TLA-modified binders before and after the RTFOT aging: complex modulus (**top**) and phase angle (**bottom**) at three specified temperatures: 40 °C, 50 °C, and 60 °C.

**Figure 4 materials-14-02532-f004:**
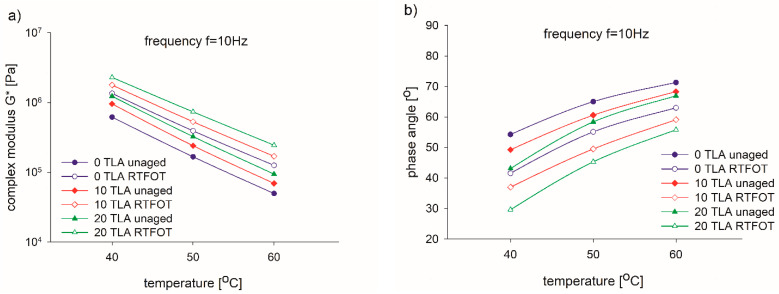
Isochronous curves of the complex modulus (**a**) and phase angle (**b**) of the TLA-modified binders before and after aging.

**Figure 5 materials-14-02532-f005:**
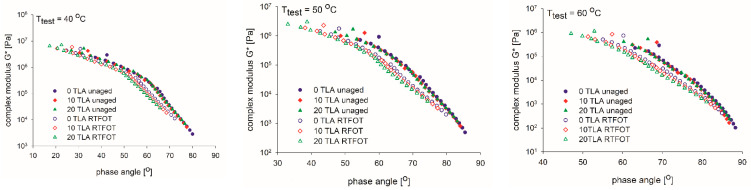
Black curves of the TLA-modified binders before and after the RTFOT aging at the specified test temperature.

**Figure 6 materials-14-02532-f006:**
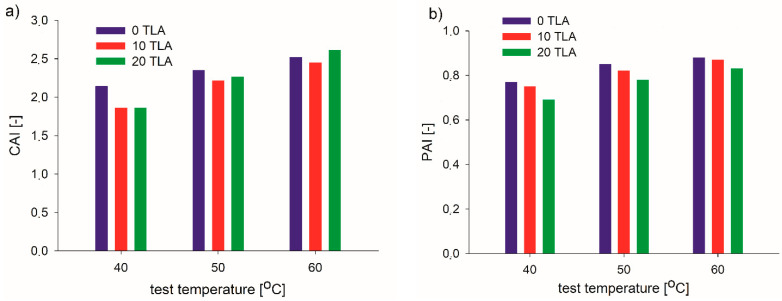
Rheological aging indices of the TLA-modified binders: (**a**) CAI (10 Hz frequency) and (**b**) PAI.

**Figure 7 materials-14-02532-f007:**
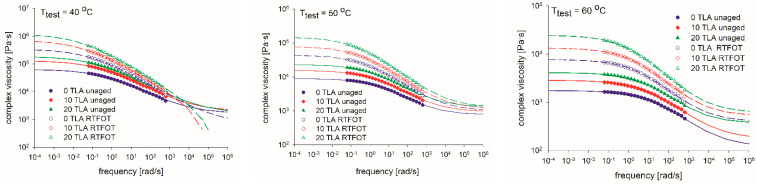
Measured and calculated viscosities versus the angular frequency on a log-log plot before and after aging at the specified test temperature.

**Figure 8 materials-14-02532-f008:**
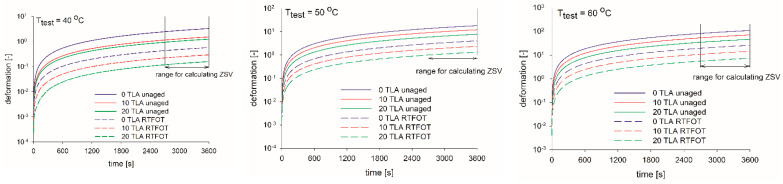
Creep curves for the TLA modified binders before and after aging at the specified test temperature.

**Figure 9 materials-14-02532-f009:**
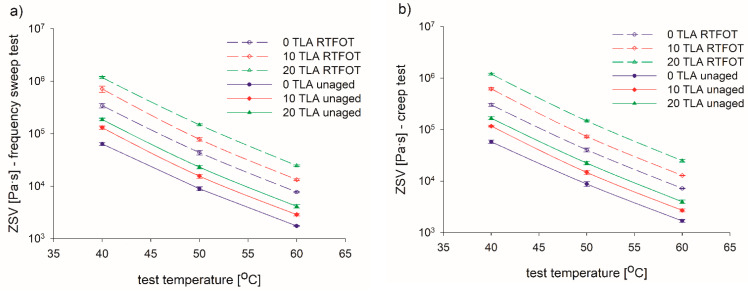
ZSV values of the tested binders from (**a**) frequency sweep test (**b**) and creep test models for various TLA contents and test temperatures before and after aging.

**Figure 10 materials-14-02532-f010:**
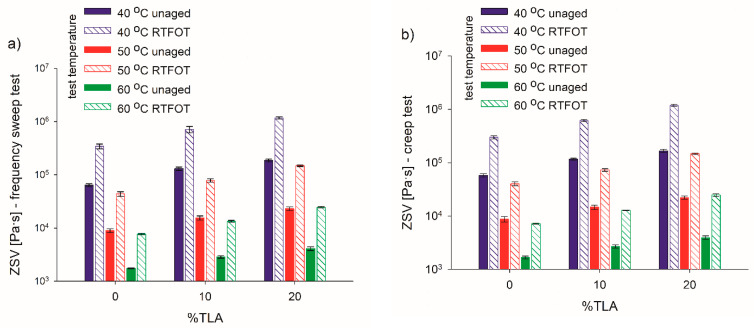
ZSV values of tested binders from the (**a**) frequency sweep test and (**b**) creep test models for various TLA content and test temperatures before and after aging.

**Figure 11 materials-14-02532-f011:**
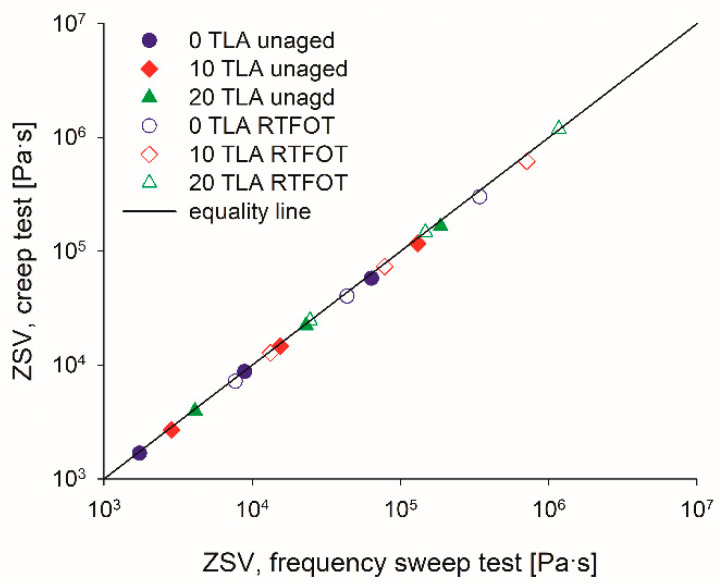
Comparison of the ZSV values of the tested binders from creep test and Cross model before and after aging.

**Figure 12 materials-14-02532-f012:**
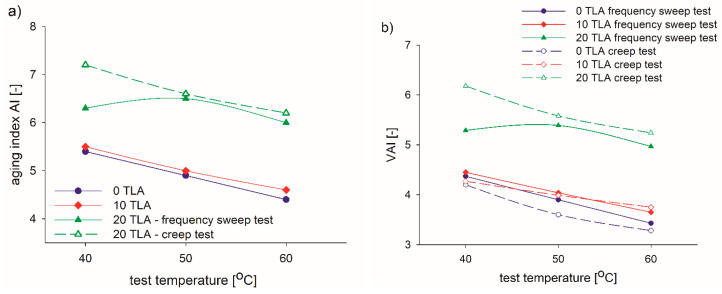
Viscosity aging indices for the TLA-modified binders: (**a**) AI and (**b**) VAI.

**Table 1 materials-14-02532-t001:** Basic properties of the 35/50 penetration-grade bitumen.

Properties	Unit	Test Methods	Test Results	Requirements According to [28]
Penetration	×0.1 mm	[29]	42.8 ± 0.6	35.0–50.0
Softening point	°C	[30]	55.1 ± 0.6	50.0–58.0
Fraass breaking point	°C	[31]	−13 ± 1.5	≤−5
Density at 25 °C	kg/m^3^	[32]	1020 ± 6	No requirements

**Table 2 materials-14-02532-t002:** Basic properties of TLA.

Properties	Unit	Test Methods	Test Results	Requirements According to [33]
Penetration	×0.1 mm	[29]	4.0 ± 0.5	0.0–4.0
Softening point	°C	[30]	101.2 ± 1.0	93.0–99.0
Solubility	% (m/m)	[34]	57.6 ± 1.0	52.0–55.0
Density at 25 °C	kg/m^3^	[32]	1380 ± 8	1390–1420

**Table 3 materials-14-02532-t003:** Physical properties of the TLA-modified binders before and after aging.

Properties	Unit	35/50 (Base Bitumen)	35/50 +10% TLA	35/50 +20% TLA
Before aging				
Penetration at 25 °C	(×0.1 mm)	42.8 ± 0.8	39.9 ± 1.2	31.8 ± 1.9
Softening point	(°C)	55.1 ± 0.6	56.9 ± 0.5	59.3 ± 0.2
After aging				
Penetration at 25 °C	(×0.1 mm)	31.5 ± 1.1	26.8 ± 1.2	22.3 ± 0.5
Softening point	(°C)	61.7 ± 0.2	65.3 ± 0.3	67.5 ± 0.5
Aging indices				
Retained penetration (RP)	(%)	73.6 ± 2.9	67.2 ± 3.6	70.1 ± 4.5
Softening point increment (SPI)	(°C)	6.6 ± 0.6	8.4 ± 0.6	8.2 ± 0.5

**Table 4 materials-14-02532-t004:** Complex modulus and phase angle of the TLA-modified binders before and after the RTFOT aging (at 10 Hz)

TLA Addition	Complex Modulus G* (MPa)			Phase Angle δ (°)		
	Test Temperature (°C)			Test Temperature (°C)		
	40	50	60	40	50	60
Before aging						
0% TLA	0.618 ± 0.036	0.167 ± 0.015	0.050 ± 0.001	54.3 ± 0.3	65.0 ± 0.2	71.3 ± 0.3
10% TLA	0.958 ± 0.060	0.240 ± 0.010	0.069 ± 0.002	49.3 ± 0.5	60.6 ± 0.5	68.3 ± 0.3
20% TLA	1.231 ± 0.043	0.325 ± 0.021	0.094 ± 0.007	43.1 ± 0.4	58.4 ± 0.4	66.9 ± 0.2
After aging						
0% TLA	1.321 ± 0.072	0.392 ± 0.032	0.126 ± 0.003	41.6 ± 0.8	55.1 ± 0.4	63.0 ± 0.6
10% TLA	1.780 ± 0.137	0.530 ± 0.031	0.169 ± 0.006	37.0 ± 0.9	49.5 ± 0.4	59.1 ± 0.2
20% TLA	2.292 ± 0.055	0.735 ± 0.017	0.245 ± 0.006	29.6 ± 0.2	45.3 ± 0.3	55.8 ± 0.2
	Complex modulus aging index (CAI)			Phase angle aging index (PAI)		
0% TLA	2.14 ± 0.17	2.35 ± 0.28	2.52 ± 0.08	0.77 ± 0.02	0.85 ± 0.01	0.88 ± 0.01
10% TLA	1.86 ± 0.18	2.21 ± 0.16	2.45 ± 0.11	0.75 ± 0.02	0.82 ± 0.01	0.87 ± 0.005
20% TLA	1.86 ± 0.08	2.26 ± 0.16	2.61 ± 0.20	0.69 ± 0.01	0.78 ± 0.01	0.83 ± 0.004

**Table 5 materials-14-02532-t005:** Zero shear viscosity of TLA-modified binders before and after aging.

TLA Addition	ZSV (Pa·s) from Frequency Sweep Test			ZSV (Pa·s) from Creep Test		
	Test Temperature (°C)			Test Temperature (°C)		
	40	50	60	40	50	60
Before aging						
0% TLA	63,840 ± 4210	8916 ± 763	1740 ± 36	57,832 ± 4089	8769 ± 1009	1682 ± 105
10% TLA	130,367 ± 9649	15,510 ± 1376	2865 ± 165	117,026 ± 4216	14,711 ± 1400	2698 ± 172
20% TLA	186,383 ± 11,031	23,063 ± 1903	4118 ± 318	166,109 ± 11,294	22,312 ± 1597	3965 ± 295
After aging						
0% TLA	342,650 ± 35,413	43,690 ± 4314	7703 ± 239	300,743 ± 20,527	40,381 ± 3619	7198 ± 125
10% TLA	710,933 ± 99,477	78,232 ± 5870	13,312 ± 568	616,827 ± 29,599	73,351 ± 4283	12,817 ± 79
20% TLA	1172,167 ± 60,001	147,400 ± 5247	24,572 ± 747	1192,585 ± 54,268	146,889 ± 6059	24,749 ± 1378
Viscosity aging index (VAI)						
0% TLA	4.37	3.90	3.43	4.20	3.60	3.28
10% TLA	4.45	4.04	3.65	4.27	3.99	3.75
20% TLA	5.29	5.39	4.97	6.18	5.58	5.24
Aging index (AI)						
0% TLA	5.37	4.90	4.43	5.20	4.60	4.28
10% TLA	5.45	5.04	4.65	5.27	4.99	4.75
20% TLA	6.29	6.39	5.97	7.18	6.58	6.24

**Table 6 materials-14-02532-t006:** Cross parameters of the TLA-modified binders before and after aging.

TLA Addition	Test Temperature (°C)	Cross Parameters			
		$\eta_{0}\;(\mathrm{ZSV})\;(\mathrm{Pa}\cdot s)$	$\eta_{\infty }\;(\mathrm{Pa}\cdot s)$	K (s)	m (-)
Before aging					
0% TLA	40	63,840 ± 4210	1675 ± 172	1.657 ± 0.162	0.405 ± 0.016
	50	8916 ± 763	773.2 ± 157.0	0.209 ± 0.007	0.462 ± 0.010
	60	1740 ± 36	177.6 ± 84.7	0.041 ± 0.005	0.459 ± 0.043
10% TLA	40	130,367 ± 9649	2035 ± 418	3.542 ± 0.348	0.405 ± 0.009
	50	15,510 ± 1376	939.1 ± 42.0	0.383 ± 0.053	0.440 ± 0.005
	60	2865 ± 165	174.6 ± 72.8	0.068 ± 0.007	0.427 ± 0.026
20% TLA	40	186,383 ± 11,031	1703 ± 486	5.148 ± 0.527	0.393 ± 0.011
	50	23,063 ± 1903	1310 ± 120	0.537 ± 0.055	0.439 ± 0.005
	60	4118 ± 318	368.3 ± 68.8	0.099 ± 0.004	0.455 ± 0.009
After aging					
0% TLA	40	342,650 ± 35,413	735.0 ± 204.4	13.676 ± 2.532	0.411 ± 0.012
	50	43,690 ± 4314	995.6 ± 70.7	1.484 ± 0.121	0.436 ± 0.004
	60	7703 ± 239	396.0 ± 21.5	0.255 ± 0.018	0.454 ± 0.003
10% TLA	40	710,933 ± 99,477	−1474 ± 820	34.439 ± 8.327	0.412 ± 0.012
	50	78,232 ± 5870	1132 ± 73	2.831 ± 0.185	0.438 ± 0.005
	60	13,312 ± 568	523.4 ± 14.0	0.479 ± 0.027	0.456 ± 0.004
20% TLA	40	1172,167 ± 60,001	−1243 ± 614	46.226 ± 7.841	0.427 ± 0.006
	50	147,400 ± 5247	1263 ± 206	5.237 ± 0.321	0.445 ± 0.004
	60	24,572 ± 747	607.5 ± 47.5	0.928 ± 0.040	0.457 ± 0.004

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
