# Peer review of "Effects of Aging on the Physical and Rheological Properties of Trinidad Lake Asphalt Modified Bitumen"

_materials, 2021, doi:10.3390/ma14102532_

Round 1
Reviewer 1 Report
I thank the Authors for their answers, which clarify the doubts raised (in the previous “Materials-1129379”).
I just advise to improve the following:
- Line 75: you wrote that “There are only a few research on aging properties of TLA-modified bitumen” but you included only 3 references from 2019 and 1 from 2020. “research” or “researches”?
- Line 102: insert a comma after “Thus”;
- Line 172 and others: please replace “75±1 min” by “(75±1) min”;
- Line 336: did you want to write “Figure 3 to 6” in “Table 4 and Figures 3 to 6 present results of …”?
- Line 529 and others: please replace “40oC” by “40 ºC”;
- You can also include some information about drawbacks or disadvantages of using this modifier.
Author Response
The author's reply to the review report of Reviewer 1 is attached as a Word file.

Reviewer 2 Report
The paper focuses on the effect of short-term aging (RTFOT aging) on the physical and rheological properties of a base bitumen modified with two different percentages of Trinidad Lake Asphalt (TLA).
The authors state that these binders will be used in mastic asphalt (MA) mixtures for bridge pavements. They also state that fatigue cracking is not important for MA pavements, for which the rutting issues are more critical. I think this induces several questionable considerations throughout the paper. It seems like the authors conclude that aging is somehow beneficial and the binder which is actually the most susceptible to short-term aging is the best one. These aspects should be carefully revised. Please, find specific comments below:
- Line 72: “On the other hand, aging is not necessarily a negative phenomenon since the resistance of the asphalt mixture to permanent deformation and the load-bearing capacity is improved due to increased stiffness and cohesion”. This reviewer disagrees with this sentence. In general, aging should be considered a degradation process of the material. A material with high aging susceptibility significantly changes its chemical and mechanical properties over time and this could lead to unexpected behaviour as compared to the conditions investigated in the lab for instance. An ideal material not affected by aging would certainly provide more predictable performance during its service life.
- Abstract: “The tests revealed that the short term aging of TLA-modified binders can improve the pavement resistance to permanent deformation and increase its load bearing capacity”. According to this reviewer, this sentence is misleading for the above-mentioned reasons
- Lines 321-326: “….the samples of 10% TLA and 20% TLA binders show almost the same anti-aging level…”, “It confirms that the 20% TLA content seems to be optimal concerning aging performance of base bitumen”. These two sentences are in contrast. The two binders have similar anti-aging performance, therefore the one with the lower TLA amount should be preferred (as stated by the authors, TLA is a very expensive material)
- Lines 389-402: “Considering the results after RTFOT aging, both TLA-modified binders exhibit smaller CAI and PAI within the entire sweep temperature range in comparison with base bitumen, indicating the better resistance to short term aging”, “Concerning PAI, the bigger TLA content, the smaller PAI, i.e. higher aging resistance, was observed”, “the 20% TLA-modified binder shows better short term aging resistance, followed by 10% TLA binder”. Smaller PAI means WORSE aging resistance (as stated also by the authors, line 275). Therefore, once again the aging resistance of the modified binders is worse than that of the base bitumen in terms of PAI, especially for 20% TLA (for which even the CAI at 60°C is higher than the base bitumen). It is not true that the binder with 20% TLA has better short-term aging resistance than the binder with 10% TLA and the base bitumen.
- Line 458: “It reveals that RTFOT aging improves TLA modification of base bitumen in terms of the viscosity stiffening effect of TLA”. Unclear sentence
- Line 512, conclusions: “The 20% TLA content seems to be optimal concerning aging performance of base bitumen”. For all the above-mentioned reasons, the reviewer does not agree with this conclusion. Overall, the results suggest that the binder with 20% TLA is the most susceptible to short-term aging
Additional comments can be found below:
- Line 64: showed instead of shown
- Line 68: fatigue cracking is considered predominant at intermediate temperatures, not at low temperatures (at which thermal cracking usually prevails)
- Line 90: “…demonstrated that, AS the TLA content and the aging time increased, the binder…”
- Line 226: “The tests were performed on smooth tiles, they were not additionally roughened. Before testing, the tiles were heated to a temperature of 70-80 °C to soften the binder and ensure good adhesion to the tiles”. What do you mean with “tiles”? Unclear
- Line 286: RTFOT instead of RTFOR
- Tables 3-4-5 are unnecessary since the same data are presented in the following figures
- All figures are very hard to read because the writings are too small. Please, modify all figures to make them readable
- It should be clearly specified also in the text that CAI and PAI are calculated at 10 Hz
- Figure 4, caption: these curves are usually called “isochronous curves”
- Line 383: Black (capital B)
- Figures 9a and 11: the legend has some typos
- Figure 9 is not even commented in the paper. Therefore I wonder if this figure is truly necessary
- Line 456: Figure 12? Probably the authors refer to Figure 10 here
- References: the references in Polish are not accessible to everyone and probably should be removed. As for the European standards, for the same reason I would suggest to mention the international EN version rather than the Polish one
Author Response
The author's reply to the review report of Reviewer 2 is attached as a Word file.

Reviewer 3 Report
This paper analyses the feasibility of suing the natural asphalt called Trinidad Lake Asphalt (TLA) as a bitumen modifier, with especial emphasis on the ageing properties.
As a general commented I would say that even though the use of this additive in bitumen modification presents a limited novelty, the topic presents evident applicative interest.
The experimental work, mainly based on rheological measurements, is very well conceived and conducted, data interpretation is reliable, referencing is appropriate. I support publication of this manuscript.
Minor comments:
- I agree that the use of TLA greatly improves the high in-service properties of bitumen, and enhanced rutting resistance is expected. However, the main concern I found is the probable deterioration of the low temperature characteristics. Then, most probably, TLA would give rise to a much harder bitumen at low temperatures, and therefore a high tendency to undergo thermal cracking. This issue deserves to be mentioned and discussed along the paper.
- Fig 7 , Sisko model curves are extrapolated at low and high frequencies and therefore out of the experimental window
Author Response
The author's reply to the review report of Reviewer 3 is attached as a Word file.

Round 2
Reviewer 2 Report
Not all the required changes have been made by the authors:
- Abstract, line 22: the authors modified this part as "...is not necessarily a negative phenomenon since the resistance of the asphalt mixture to permanent deformation and the load-bearing capacity are not reduced due to non-worsened stiffness after aging". However, in the answer to the reviewer they wrote that this part was modified as "The tests revealed that the short term aging of TLA-modified binders does not worsen or reduce the pavement resistance to permanent deformation and the load bearing capacity of the asphalt mixture”. This second sentence is probably better
- The authors stated that they modified all the figures to make them more readable. However, the figures seem to be exactly the same as in the previous version of the manuscript, and therefore the writings are still too small to be read
- Lines 485-498: the authors modified this part very summarily, leading to several inconsistencies. Line 488: I would suggest to delete "indicating the smaller resistance to short term aging". Line 489: I would suggest to delete "positive". Line 493: "Concerning PAI, the bigger TLA content, the smaller PAI, i.e. higher aging resistance, was observed, regardless test temperature". Smaller PAI means worse aging resistance, as already underlined by this reviewer in the previous review. Line 495: "As for CAI, PAI shows that the aging influence is more visible in higher test temperature". Is it CAI or PAI here? Line 496: "Finally, through synthetical consideration of CAI and PAI, the 20% TLA-modified binder shows better short term aging resistance, followed by 10% TLA binder". In the answer to the reviewer, the authors stated that they modified this part as "Considering all of the analyzed properties (RP, SPI, CAI and PAI) it can be observed, that for 20% of TLA addition three of four properties are better than for 10% addition (CAI and PAI as the average values for temperatures 40, 50 and 60°C)”, which would be more appropriate.
Additional comments:
- Line 51: the authors added that TLA modified binders are less resistant to low temperature cracking (which is reasonable given their stiffening effect on the binder). However, few lines below (line 59) they state that TLA improves low temperature cracking characteristics. Please, modify line 59
- Line 58-59: anti-rutting performance and resistance to high temperature deformation are the same thing
- line 86: "on" should be deleted
- line 186: "whose" instead of "which"
- line 202: "it" instead of "them"
- G* and δ are mentioned many times in the main sentences in parentheses. These parentheses should be removed (for instance, see line 433 and line 457)
- "regardless" always goes with "of" (please, change it throughout the paper)
Author Response
The authors are very grateful once again to Reviewer 2 for valuable comments and editorial corrections. All minor changes and revisions suggested by the Reviewer are included in the text and clearly highlighted using the "Track Changes" function. Since there are mostly typographical errors, no further detailing of changes seems to be necessary. Moreover, all the figures were modified once again to make them more readable.
This manuscript is a resubmission of an earlier submission. The following is a list of the peer review reports and author responses from that submission.
Round 1
Reviewer 1 Report
The present manuscript assesses the effect of short-term aging and on the physical and rheological properties of Trinidad Lake Asphalt (TLA)-modified bitumen. The manuscript is very well written (although some English errors should be corrected), the experiments are well designed and the objective seems interesting based on a novelty subject with potential applications in the asphalt industry. However, revision is required and some comments should be taken into account:
- Introduction. Some references should be included. For example, lines 60-62: “Results from many studies…” or lines 121-127 in order to differentiate the effect of short-term and long-term aging influence on the mechanical properties.
- Introduction. The objective is twice mentioned (lines 93 and 115).
- Introduction. The explication about the 35/50 bitumen used and the different TLA content (lines 103-114) should be included in the Materials and Methods section.
- Materials and Methods. What is the resting time before performing the aging procedure and after preparation?
- The effect of ageing on bitumen chemistry could be studied but, at least, the integration of TLA in the base bitumen must be assessed by microscopy and included in the manuscript.
- Rheological measurements. Have rough surfaces been used to prevent slippage? Has a linear viscoelastic interval determination been performed with strain/stress sweep tests?
- Please, clarify that AI and VAI have been used with complex and not complex viscosity.
- I miss a means comparison test to assess whether or not the differences are significant.
- “Figure 4” must in bold in the caption.
- Figure 12b: “Sweep test” in legend, do the authors mean “frequency sweep tests”?
- Of all the systems studied, which one would the authors select as the best? A response surface methodology study would be welcome.
Reviewer 2 Report
- In the last line of the abstract, you should include some major conclusion from your work.
- The language should be improved.
- Line 26-27: Isn't it because of higher traffic?
- What do you mean by superior rheological properties?
- Table 2 (and lines 138-139): TLA has NOT fulfilled requirements.
- Line 151: This information is not necessary.
- You need more consistent and conclusions. Don`t repeat your results or discussion.
- The manuscript does not provide any new finding. I could probably write the conclusions without doing any tests.
Reviewer 3 Report
Congratulations to authors. Very interesting and complete paper about the effect of short-term aging on the physical and rheological characteristics of a 35/50 penetration grade bitumen modified by two different Trinidad Lake asphalt (TLA) contents.
In general, the manuscript is well presented but some information shall be clarified or completed:
- Line 129: did the Authors have any information about the TLA-modified binder storage stability? Taking into account the differences in density (1020 ± 6 kg/m3 for 35/50 and 1380 ± 8 kg/m3 for TLA-modified binder), some information regarding this topic will be welcome;
- Tables 1 and 2: please replace the unit of penetration (⨯0.1 mm instead of x0,1 mm);
- Lines 135/139: this sentence shall be placed before Tables 1 and 2;
- Lines 146/147: why did you use such mixing conditions (3000 rpm for 5 min @ 160 °C)? I advise the Authors to justify this option;
- Table 3: in the base bitumen (35/50), are the penetration and softening point exactly the same before and after aging (42.8±0.8 and 55.1±0.6)?
- Line 216: please include the software version for the “HAAKE RheoWin”;
- In “Conclusion” maybe some graphical representation of the trends will help readers;
- This research was developed by simulating only the short-term aging for both binders. Do you already have any inferences about the performance of TLA-modified binder at long-term (during its service life)?
Round 2
Reviewer 1 Report
In the revised manuscript, authors made some necessary revision and basically addressed the responses to the questions my comments. The paper was generally improved. However, I suggest clarifying some points before consideration of the manuscript:
-Please, include in the manuscript the description about rheological measurements commented in the response.
- Statistical section should be included in the Experimental section as well as the mean comparison tests with different letters to assess significant differences (not only error lines).
Author Response
Please find attached response.

Reviewer 2 Report
.
Author Response
Thank you very much for the subsequent comments and suggestions. We have done our best to answer and improve our text strictly according to the Reviewer’s remarks. Particularly, we have checked English language and style, improved the introduction for sufficient background and included all relevant references. We have also revised the conclusions to be supported by the results.
Round 3
Reviewer 1 Report
In the revised manuscript, the authors made some necessary revision and basically addressed the responses to the questions my comments. The paper was generally improved and, in my opinion, can be published.